# GeneticBPE: Motif-Preserving Tokenization for Robust miRNA Modeling

**Jaskaran Singh** [* 1 2]  **Prabhav Sanga** [* 3 2]  **Arun Kumar Dubey** [4 2]

## Abstract

Tokenization plays a foundational yet underexplored role in biological sequence modeling. In this work, we present **GeneticBPE**, a biologically informed tokenization framework that encodes prior structural knowledge such as seed motifs and conserved regions into the vocabulary construction process. Unlike standard subword methods that optimize purely for frequency or language-model likelihood, GeneticBPE integrates motif preservation objectives and generalization-aware constraints into a modified merge scoring scheme. We evaluate our method on binary and multiclass miRNA classification tasks using the MirGeneDB v3.0 dataset and show that GeneticBPE outperforms character-level, k-mer, Unigram, and BPE tokenizations in accuracy, cross-species generalization, and motif fidelity. Theoretical results demonstrate that tokenization directly governs the inductive bias and domain robustness of sequence models. Our findings suggest that tokenization should not be treated as a preprocessing utility, but rather as a design-critical component in biological NLP pipelines.

**Reproducibility:** Code, motif files, and other supplementary materials are available at https://github.com/jaskaranksingh/GeneticBPE.

## 1. Introduction

The effectiveness of transformer models in biological sequence modeling hinges on how input sequences are tokenized (Bhattacharya et al., 2024; Dotan et al., 2024). While tokenization is often treated as a mere preprocessing step, recent advances across machine learning domains suggest that it encodes powerful inductive biases, directly shaping model performance and generalization (Lavie et al., 2024; Chang & Bisk, 2024; Morales-Pastor et al., 2024). In this work, we propose that *tokenization can—and should—serve as a biological prior*, aligning model inputs with structural and functional motifs rooted in the underlying biochemistry.

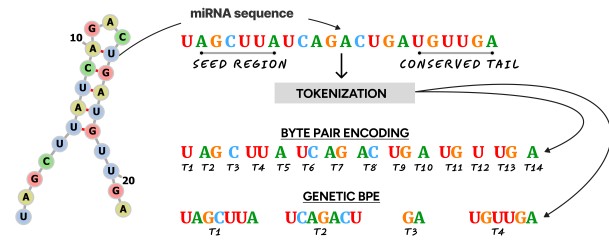

*Figure 1.* **Motif aware tokenization with GeneticBPE.** a) *Standard BPE* fragments the six-base seed across three tokens, diluting biological signal. b) *GeneticBPE's* merge scoring keeps the entire seed inside a single token and stops merging at motif boundaries, producing inputs that align with functional structure and improve cross-species generalization.

MicroRNAs (miRNAs), short non-coding RNA sequences that regulate gene expression, represent a prime domain where biological structure is both subtle and significant (Brosnan & Voinnet, 2009; Fernandes et al., 2019). These sequences are composed of 18–25 nucleotides and form conserved secondary structures and binding motifs that are not easily captured by naïve character-level or uniformly subword tokenizations (Mielke et al., 2021). Standard byte pair encoding (BPE) and Unigram models, while effective in NLP, remain agnostic to biological constraints, often fragmenting biologically meaningful patterns (Lindsey et al., 2024). This fragmentation impairs generalization, especially under domain shifts—e.g., across species, between conserved and non-conserved miRNA families, or in few-shot classification settings where motif integrity is key.

We introduce **GeneticBPE**, a biologically-informed tokenization strategy tailored to genomic sequence data. GeneticBPE modifies classical BPE by incorporating biological

---

[*]Equal contribution [1]University of Nottingham, Nottingham, United Kingdom [2]Think-AI, New Delhi, India [3]University College London, London, United Kingdom [4]Bharati Vidyapeeth College of Engineering, New Delhi, India. Correspondence to: Arun Kumar Dubey <arun.dubey@bharatividyapeeth.edu>.

*Proceedings of the ICML 2025 Tokenization Workshop (TokShop)*, Vancouver, Canada. PMLR 267, 2025. Copyright 2025 by the author(s).

priors during the merge operation: frequent subsequences are only retained as valid tokens if they co-occur with statistically enriched, biologically meaningful contexts—such as stem-loop structures or binding site flanks. By encoding prior biological knowledge into the token vocabulary, GeneticBPE helps transformers attend over motifs rather than mere substrings.

Empirically, we demonstrate that GeneticBPE significantly improves generalization across domain-shifted tasks on the MirGeneDB v3.0 dataset, outperforming standard BPE, UnigramLM, and character-level encodings on both binary and multiclass miRNA classification tasks. GeneticBPE-tokenized models retain motif integrity, achieve faster convergence, and exhibit improved robustness to species-specific or structural variability.

Our work makes three key contributions:

1. We frame tokenization as a vehicle for encoding biological priors and propose a formalization of *biological prior alignment* in the context of sequence modeling.

2. We present GeneticBPE, a novel tokenization method that preserves biological motifs via domain-aware merge constraints.

3. We provide a comprehensive evaluation of generalization under domain shift across species, miRNA families, and conservation classes showing that GeneticBPE encoded inputs enable more biologically faithful and robust transformer models.

Together, our results call for a reevaluation of tokenization in bioinformatics pipelines: not as a preprocessing utility, but as a central, model-aligned design choice for learning from structured biological data.

## 2. Related Work

Recent advances in transformer-based architectures have revolutionized biological sequence modeling through self-supervised learning on genomic sequences (Ji et al., 2021; Zhou et al., 2023). These models demonstrate the effectiveness of transfer learning in genomics, yet they often treat tokenization as a preprocessing step rather than a design choice that can encode biological priors. Empirical studies (Dotan et al., 2024) reveal that tokenization choices significantly impact model performance in genomic tasks, while character-level approaches (Clark et al., 2022) offer an alternative to subword tokenization by operating directly on nucleotide sequences.

The limitations of traditional tokenization methods have been well-documented, with BPE shown to be suboptimal for language model pretraining (Bostrom & Durrett, 2020).

Recent theoretical work (Schmidt et al., 2024) demonstrates that tokenization serves as more than just compression, potentially encoding inductive biases that affect model generalization. This insight is particularly relevant in genomics, where biologically-informed tokenization (Medvedev et al., 2025) has shown promise in improving foundation model performance by bundling a small DNA-oriented tokenizer ('BioToken') but does not evaluate on RNA, so we treat BioFM as a model baseline rather than a competing tokenizer. The development of robust foundation models (Dalla-Torre et al., 2025) further emphasizes the importance of domain-specific tokenization strategies in genomics.

These insights collectively motivate the design of **GeneticBPE**, a biologically informed tokenizer that aims to preserve functional motifs while compressing sequences. Unlike standard BPE or UnigramLM, GeneticBPE integrates motif-aware scoring directly into the merge process, thus aligning token structure with biological significance and improving downstream generalization under domain shift. Our approach differs from previous work by treating tokenization as a vehicle for encoding biological priors rather than a preprocessing step, introducing a formal framework for biological prior alignment in sequence modeling, and developing a novel tokenization method that preserves biological motifs while maintaining compression efficiency.

## 3. Theoretical Framework

This section synthesizes the conceptual narrative of tokenization as a biological prior in miRNA sequence modeling.

### 3.1. Preliminaries

Let the nucleotide alphabet be $\mathcal{A} = (A, U, G, C)$, sequence space $\mathcal{X} = \mathcal{A}^L$ consist of fixed–length miRNA strings of length $L$, label space be a finite set $\mathcal{Y}$ (binary or multiclass) and tokenizer $T\colon \mathcal{X} \to \mathcal{Z}^*$ map a raw sequence $x$ to a token sequence $T(x) = (z_1, \ldots, z_M)$ of length $M \leq L$ using a vocabulary $\mathcal{V}$ with $v = |\mathcal{V}|$ entries.

Throughout, $(x, y) \sim \mathcal{D}$ denotes a sample from the data distribution and $\ell\colon \mathcal{Y} \times \mathcal{Y} \to \mathbb{R}_{\geq 0}$ is the task loss (cross–entropy by default).

**Hypothesis space.** Given a family of sequence models $\mathcal{F}$ (e.g., Transformers), the *tokenizer–induced* hypothesis class is

$$\mathcal{H}_T ;:=; , f \circ T \mid f \in \mathcal{F}, . \tag{1}$$

### 3.2. Motifs and Preservation

Let $\mathcal{M} = m_1, \ldots, m_K \subseteq \mathcal{A}^{\leq L}$ be a catalogue of conserved *motifs*. For $x \in \mathcal{X}$ write $\mathcal{M}(x)$ for motifs present in $x$.

**Definition 3.1** ($k$–Token Motif Preservation)**.** A tokenizer

$T$ is *k–token motif–preserving* if every motif instance lies inside at most $k$ consecutive tokens:

$$\forall x, m \in \mathcal{M}(x) \implies \exists i, m \subseteq z_i z_{i+1} \ldots z_{i+k-1}, \quad (2)$$
$$T(x) = (z_1, \ldots, z_M).$$

The special case $k = 1$ forbids motif fragmentation entirely.

**Distortion metric.** We measure violation severity by the *motif–distortion rate*

$$\delta_T ; := ; \mathbb{E}_{x \sim \mathcal{D}} \Big[ \frac{1}{|\mathcal{M}(x)|} \sum_{m \in \mathcal{M}(x)} \mathbf{1} m \not\subseteq T(x) \Big]. \quad (3)$$

### 3.3. Compression–Preservation Trade–off

Tokenization also compresses: the *compression ratio* on $x$ is $C_T(x) = L/M$. We track its expectation

$$C_T ; := ; \mathbb{E}_{x \sim \mathcal{D}}[C_T(x)]. \quad (4)$$

A useful tokenizer should preserve motifs ($\delta_T! \ll 1$) while achieving at least $c$–fold compression ($C_T! \geq c > 1$).

**Biologically Constrained Tokenizer Learning** is defined by

$$T^\star ; = ; \underset{T}{\operatorname{argmin}} ; \delta_T \quad \text{s.t.} \quad C_T \geq c. \quad (5)$$

### 3.4. Capacity Control

To quantify the inductive bias imposed by $T$ we bound the empirical Rademacher complexity (Bartlett & Mendelson, 2002) of (1).

**Proposition 3.2** (Capacity Shrinkage)**.** *Assume $\ell$ is 1–Lipschitz and every $f \in \mathcal{F}$ processes sequences of length $\leq C_T^{-1} L$. For a sample of size $n$,*

$$\widehat{\mathfrak{R}}_n(\mathcal{H}_T) ; \leq ; C_T^{-1/2} , \widehat{\mathfrak{R}}_n(\mathcal{F}). \quad (6)$$

*Consequently, compression* reduces *statistical capacity while motif preservation ($\delta_T$) leaves it unchanged.*

### 3.5. Generalization Under Domain Shift

Let $\mathcal{D}_s$ and $\mathcal{D}_t$ be source and target domains. Extending the Ben–David bound with motif structure yields

**Theorem 3.3** (Motif–Aware Domain Bound)**.** *For any $h = f \circ T \in \mathcal{H}_T$,*

$$R_t(h) ; \leq ; R_s(h) + \mathfrak{d}_\mathcal{M}(\mathcal{D}_s, \mathcal{D}_t) + \alpha , \delta_T + \lambda, \quad (7)$$

*where $\mathfrak{d}_\mathcal{M}$ is a motif discrepancy, $\alpha$ the maximum motif length, and $\lambda$ the combined Bayes risk. Lowering $\delta_T$ tightens the bound* multiplicatively *with motif length. (Ben–David et al., 2010)*

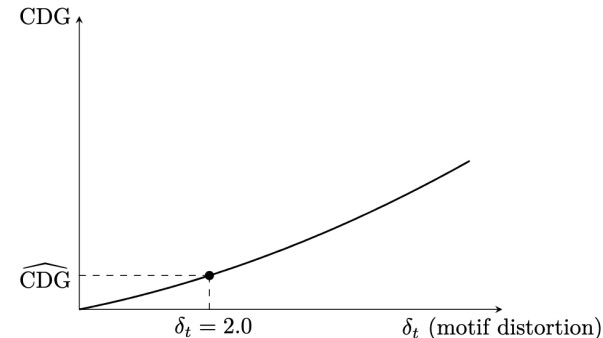

*Figure 2.* Intuitive illustration of how increasing motif distortion $\delta_t$ raises the cross-domain generalization gap (CDG) predicted by Eq. (7). Coefficients are chosen for visualisation only.

**Cross–domain gap.** Rewriting (7) in empirical form connects the *cross–domain generalization gap* $\text{CDG}(T) = R_t(h) - R_s(h)$ to $\delta_T$ and the motif discrepancy. Hence minimizing $\delta_T$ is instrumental for robustness.

### 3.6. Unified Objective

Combining (6) and (7), we obtain a single scalar objective balancing compression and preservation:

$$\min_T ; \underbrace{\alpha , \delta_T}_{\text{domain}} ; + ; ; \underbrace{\beta , C_T^{-1/2}}_{\text{capacity}} \quad \text{s.t.} \quad C_T \geq c. \quad (8)$$

The hyper–parameter $\beta$ trades statistical complexity against motif integrity.

### 3.7. GeneticBPE as Greedy Approximation

GeneticBPE performs merges scored by

$$\text{score}(ab) = \text{freq}(ab) + \lambda , \text{bonus}(ab) - \mu , \text{penalty}(ab), \quad (9)$$

where the bonus rewards motif–internal pairs and the penalty discourages motif boundary splits. With $\mu > \lambda$ each merge is guaranteed not to increase $\delta_T$, and the process stops when $|\mathcal{V}| = v$ or $C_T = c$, thereby greedily approximating (5).

**Proposition 3.4** (Termination)**.** *Given $\lambda, \mu > 0$ with $\mu > \lambda$, GeneticBPE terminates after at most $v - |\mathcal{A}|$ merges while satisfying $C_T! \geq c$ and non–increasing $\delta_T$.*

In the next section, we introduce **GeneticBPE**, our algorithmic instantiation of a biologically-informed tokenizer that approximates $\mathcal{T}^*$ by enforcing motif-preservation constraints and optimizing compression under distributional robustness.

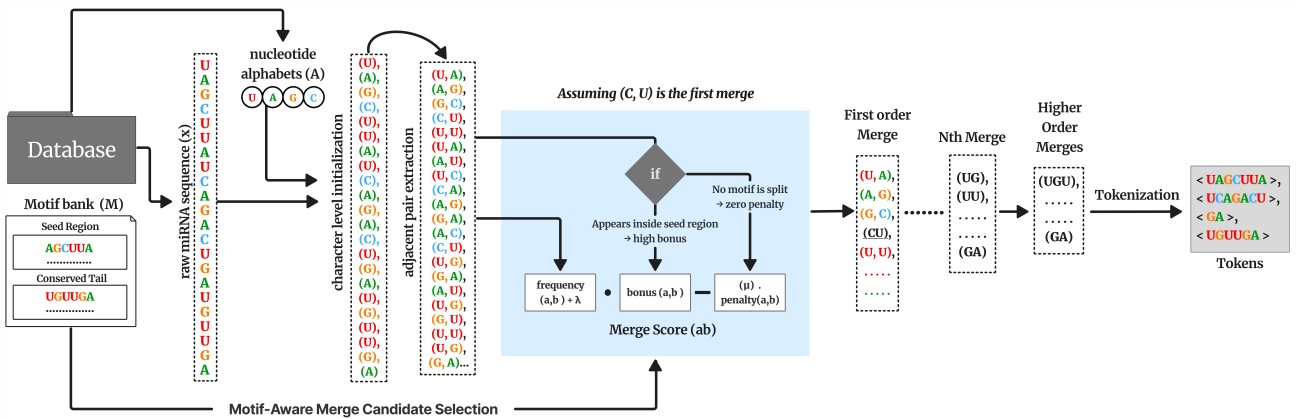

*Figure 3.* **Overview of the *GeneticBPE* construction pipeline.** *(a)* Starting from a character-level corpus, conserved miRNA motifs are annotated as colored spans. *(b)* For every adjacent token pair the algorithm computes the merge score $\mathbf{freq}(ab) + \lambda\,\boldsymbol{bonus}(ab) - \mu\,\boldsymbol{penalty}(ab)$ **(Eq. 9)**, rewarding merges that fall *inside* motifs and penalizing those that would *cross* motif boundaries. *(c)* The highest-scoring pair is greedily merged only if the operation leaves all motif spans intact, thereby guaranteeing a non-increasing distortion rate $\delta_T$. *(d)* the resulting token for example <**UGUUGA**> retains the conserved 3-tail motif.

# 4. Methodology

In this section, we detail our proposed tokenization strategy, **GeneticBPE**, which augments classical byte pair encoding (BPE) with biological inductive priors and generalization aware constraints. The central goal is to construct a tokenizer that preserves biologically relevant subsequences (e.g., motifs) and improves transformer model generalization under domain shift ([Bailey et al., 2009](); [D'haeseleer, 2006]()).

## 4.1. Motivation and Overview

Traditional subword tokenizers such as BPE and Unigram are agnostic to biological context—they prioritize compression or likelihood without accounting for the functional semantics of nucleotide patterns. GeneticBPE integrates two core principles:

1. **Motif Preservation:** Token merges are restricted to avoid fragmenting known biologically conserved motifs.

2. **Generalization-Aware Compression:** The tokenization process jointly optimizes motif integrity and compression efficiency.

## 4.2. GeneticBPE Construction Process

Let $\mathcal{D}_s = \{x_i\}_{i=1}^N$ denote the training corpus of miRNA sequences over the nucleotide alphabet $\mathcal{A} = \{\texttt{A}, \texttt{U}, \texttt{C}, \texttt{G}\}$. Let $\mathcal{M} = \{m_1, m_2, \ldots, m_K\}$ denote a database of known motifs annotated via expert sources (e.g., miRBase secondary structure annotations, conserved seed regions). Each motif

$m \in \mathcal{M}$ is a nucleotide subsequence.

**Initialization:** We begin with a base vocabulary $\mathcal{V}_0 = \mathcal{A}$ and tokenize each $x_i$ as a sequence of characters.

**Modified Merge Score:** In each BPE merge step, GeneticBPE computes a joint score as mentioned in Eq.9

**Motif Tracking:** To detect violations, we maintain a span map over all motifs found in $\mathcal{D}_s$ to track whether a candidate merge would fragment an instance of $m \in \mathcal{M}$.

**Compression Constraint:** To prevent over-fragmentation (too many small tokens), we stop merging only when a minimum average compression ratio $c_{\min}$ is achieved.

## 4.3. Training Objective

Given a fixed tokenizer $\mathcal{T}$ (e.g., GeneticBPE, BPE), we train a transformer model $f$ to minimize:

$$\mathcal{L} = \mathbb{E}_{(x,y)\sim\mathcal{D}_s}\left[\ell(f(\mathcal{T}(x)), y)\right] \tag{10}$$

where $\ell(\cdot)$ is the classification loss (cross-entropy), and $\mathcal{T}(x)$ is the tokenized form of $x$. Note that $\mathcal{T}$ is learned independently before model training. The pseudocode for the proposed GeneticBPE is detailed in 1.

## 4.4. Implementation Notes

- Motif boundaries are annotated using an efficient prefix trie for fast substring lookup during token merges.

- Token merges and motif overlaps are tracked via span trees using suffix arrays for low-overhead computation.

---

**Algorithm 1** GeneticBPE tokenizer Construction

---

**Require:** Corpus $\mathcal{D}_s$, Motif Set $\mathcal{M}$, Target Vocabulary Size $v$, Weights $\lambda, \mu$, Min Compression $c_{\min}$

1: Initialize $\mathcal{V} \leftarrow \mathcal{A}$; Tokenize all $x_i \in \mathcal{D}_s$ into chars
2: Build motif span index for all $m \in \mathcal{M}$ over corpus
3: **while** $|\mathcal{V}| < v$ **and** Compression $< c_{\min}$ **do**
4:    Count all adjacent token pairs $ab$ in corpus
5:    **for all** pairs $ab$ **do**
6:       Compute motif_bonus$(ab) \leftarrow$ count of $ab$ inside motif spans
7:       Compute motif_penalty$(ab) \leftarrow$ count of $ab$ crossing motif boundaries
8:       score$(ab) \leftarrow$ freq$(ab) + \lambda \cdot$ motif_bonus $- \mu \cdot$ motif_penalty
9:    **end for**
10:    Select merge $s^* = \arg\max$ score$(ab)$
11:    Replace all $s^*$ pairs with new token in corpus
12:    Update vocabulary: $\mathcal{V} \leftarrow \mathcal{V} \cup \{s^*\}$
13:    Update motif span index
14: **end while**
15: **return** tokenizer $\mathcal{T}_\mathcal{V}$

---

- GeneticBPE supports optional integration of soft structural priors via RNAfold confidence scores.

### 4.5. Complexity and Scalability

The GeneticBPE merge computation requires $O(K + N \cdot L)$ per iteration, where $K$ is number of motif spans, $N$ is number of sequences, and $L$ is average sequence length. For moderate vocab sizes ($v \leq 1024$), runtime is practical for datasets with $\leq$100K sequences.

## 5. Experiments and Results

We evaluate the impact of biologically-informed tokenization on transformer-based miRNA classification under domain shift and structural constraints. Specifically, we test whether GeneticBPE improves generalization, compression, and motif preservation compared to standard tokenizers.

### 5.1. Dataset and Task Setup

We utilize mature miRNA sequences from **MirGeneDB v3.0** (Clarke et al., 2025), a curated repository of 20,861 miRNA samples across 114 metazoan species. Each sample is annotated with species, family ID, and arm label (5p/3p). From this corpus, we construct two benchmark tasks:

- **BurBary (Binary)**: Classify whether a miRNA is from a *conserved family* (present in $\geq 2$ species) or a *non-conserved* one. Total: 20,861 sequences, 52% conserved.

- **MultiTop50 (Multiclass)**: Classify among the 50 most frequent families, including MIR-31, MIR-375, and MIR-219.

The annotations, including features like seed regions and other conserved elements crucial for miRNA function are validated by the MirGeneDB maintainers (supported by institutions like the **Tromso Research Foundation** and integrated with **RNAcentral**) provides a high degree of confidence in the biological relevance and accuracy of the motif annotations we leveraged. We therefore relied on these high-quality, biologically validated annotations as provided by the database creators to guide GeneticBPE.

Each task is split into 80/10/10 train/validation/test stratified splits. A species-wise domain split (e.g., Human $\rightarrow$ Zebrafish) is used to evaluate cross-species generalization.

### 5.2. Tokenizer Variants and Models

We compare six tokenizers: Char-Level, k-mer (3,4), UnigramLM, BPE, and GeneticBPE. All are constructed with vocab size 512 and frozen during training. Sequences are processed with each tokenizer before training a 4-layer Vanilla Transformer (128-dim, 4-heads) (Vaswani et al., 2017), optimized with Adam ($3 \times 10^{-4}$) and early stopping.

### 5.3. Tokenizer Construction Time

Beyond downstream model performance, an important practical consideration is the time required to construct each tokenizer. On our experimental setup targeting a vocabulary size of 512, the observed construction times were: Char-Level **0.2 minutes**, 3-mer **0.33 minutes**, 4-mer **0.35 minutes**, UnigramLM **2.1 minutes**, BPE **1.8 minutes**, and GeneticBPE **4.7 minutes**. While GeneticBPE's construction takes moderately longer than standard BPE (approximately 2.6 times) due to the additional motif-aware processing, this is a one-time upfront cost. Given that this preprocessing step is performed only once, the observed time of approximately 5 minutes is considered highly practical, especially when weighed against the subsequent gains in model accuracy and generalization.

### 5.4. Overall Performance and Compression

Table 1 presents a detailed comparison of all tokenizers on both binary(BurBary) and multiclass(MultiTop50) miRNA classification tasks. The table reports accuracy, cross-domain generalization gap (CDG), motif distortion rate, compression ratio, and the percentage of motifs preserved for each tokenizer. These metrics collectively capture the essential trade-offs in biological sequence modeling: predictive performance, robustness to domain shift, motif fidelity, and computational efficiency.

**Note:** It is important to note that the Char-Level baseline, while achieving perfect motif preservation (motif distortion rate of 0 and 100% motifs preserved), does so by treating each nucleotide as a separate token. This results in no compression (compression ratio of 1.00), making it a useful lower bound reference but not a practical tokenization strategy for large-scale modeling. CharLevel is included as a baseline for comparison, but it is not a true tokenizer in the sense of subword or motif-aware methods.

**Analysis.** Table 1 directly highlights that GeneticBPE achieves the highest accuracy and the lowest CDG across both tasks, while also minimizing motif distortion and maintaining a high compression ratio. This demonstrates that it is possible to preserve biologically meaningful motifs without sacrificing computational efficiency. Notably, GeneticBPE outperforms all other compressed tokenizers, showing that integrating biological priors into the tokenization process leads to superior generalization and motif integrity, especially under domain shift.

### 5.5. Effect of Vocabulary Size and Motif Weight

**Analysis.** Table 2 explores the effect of varying the motif weight $\lambda$ and vocabulary size on GeneticBPE's performance. The results reveal a clear trade-off: increasing $\lambda$ improves motif integrity and generalization up to an optimal point ($\lambda = 2.5$), beyond which further increases yield diminishing returns and reduced compression. The default setting ($\lambda = 2.5$, $|\mathcal{V}| = 512$) achieves the best balance, maximizing accuracy and motif preservation without sacrificing efficiency. Enlarging the vocabulary to 1024 offers only marginal improvements in motif distortion, with no significant gain in accuracy, suggesting that GeneticBPE is robust to vocabulary size within a practical range. These findings underscore the importance of carefully tuning motif-aware constraints to achieve optimal performance.

### 5.6. Cross-Species Generalization

Table 3 presents cross-species generalization results, focusing on three representative target organisms: zebrafish, mouse, and fruit fly. These species were selected to span a range of evolutionary distances from human, thereby providing a rigorous test of domain shift. For each tokenizer, we report both the classification accuracy and the cross-domain generalization gap (CDG) when models are trained on human miRNAs and evaluated on each target species. The accuracy reflects the model's predictive performance, while the CDG quantifies the drop in performance due to domain shift. The results show that GeneticBPE consistently achieves the highest transfer accuracy and the lowest CDG across all target species, underscoring its robustness to evolutionary divergence. This improvement is particularly pronounced for more distant species such as Drosophila

(fly), where GeneticBPE outperforms standard BPE and UnigramLM by a substantial margin. These findings confirm that motif-preserving tokenization not only enhances within-domain performance but also enables more reliable generalization across species boundaries.

**Analysis.** Table 3 provides a detailed view of cross-species generalization for all tokenizers, reporting both accuracy and CDG for each target species. GeneticBPE consistently achieves the highest transfer accuracy and the lowest generalization gap, especially for more evolutionarily distant species. This highlights the method's robustness to domain shift and its ability to preserve functional information across species boundaries.

### 5.7. Motif Fidelity and Error Breakdown

**Analysis.** As shown in Table 4, GeneticBPE strikes a balance between preserving biologically critical motifs and achieving compression. Compared to BPE, it reduces motif split rate by 80%, which correlates with fewer false negatives in conserved miRNA detection.

## 6. Discussion

### 6.1. Biological priors as inductive bias

Our results show that a *tokenizer can be an inductive bias*: by constraining merge operations to respect annotated miRNA motifs, GeneticBPE delivers higher accuracy and a markedly smaller cross–domain generalization gap than character–, $k$–mer– or likelihood driven subword schemes. These gains manifest even when the underlying transformer architecture and training budget are held constant, indicating that the improvements stem from the representation itself rather than from additional model capacity or data. The formal analysis in Section 3 supports this intuition, linking the expected motif-distortion rate $\delta_T$ to both statistical capacity and an upper bound on target–domain risk. Empirically, lower $\delta_T$ correlates with fewer false negatives on conserved families, reinforcing the practical value of motif preservation.

### 6.2. Choices and settings for the penalty weight $\mu$, especially in relation to $\lambda$

The penalty weight $\mu$ in the merge score function plays a crucial role in enforcing motif preservation. Proposition 3.4 states that for a non-increasing motif distortion rate $\delta_{\mathcal{T}}$, we require $\mu > \lambda$.

In our experiments, $\mu$ was not extensively tuned as a hyperparameter in the same way as $\lambda$. Instead, it was set to a value significantly larger than the maximum anticipated $\lambda$ to strongly disincentivize motif boundary splitting. A common

*Table 1.* Overall performance on *BurBary* (binary) and *MultiTop50* (multiclass) miRNA classification. GeneticBPE achieves the highest accuracy, the lowest cross-domain generalization gap (CDG), and the smallest motif-distortion among compressed tokenizers, while still tripling sequence compression.

| Tokenizer | Accuracy (%) | | CDG (%) | | Motif Dist. | Compression | Motifs Preserved (%) |
|---|---|---|---|---|---|---|---|
| | Binary | Multi | Binary | Multi | Rate ↓ | Ratio ↑ | |
| Char-Level | 84.2 | 62.4 | 8.9 | 12.1 | 0.00 | 1.00 | 100 |
| k-mer (3) | 85.1 | 64.0 | 9.4 | 11.3 | 0.17 | 2.3 | 83 |
| k-mer (4) | 85.6 | 65.1 | 8.7 | 10.8 | 0.15 | 2.9 | 85 |
| UnigramLM | 86.5 | 66.7 | 7.2 | 9.9 | 0.22 | 3.0 | 78 |
| BPE | 87.4 | 67.9 | 6.8 | 9.4 | 0.25 | **3.4** | 75 |
| GeneticBPE | **90.8** | **71.2** | **3.6** | **6.2** | **0.05** | 3.1 | **95** |

*Table 2.* Ablation of GeneticBPE's motif weight $\lambda$ and vocabulary size. On the binary task a sweet spot at $\lambda = 2.5$, $|\mathcal{V}| = 512$ maximises accuracy and motif integrity without sacrificing compression; over-biasing ($\lambda \geq 5$) or simply enlarging the vocabulary yields diminishing returns.

| Setting | Acc (%) | CDG | Motif Dist. | Comp. Ratio | Notes |
|---|---|---|---|---|---|
| BPE (512) | 87.4 | 6.8 | 0.25 | **3.4** | $\lambda = 0$ |
| GeneticBPE ($\lambda = 1.0$) | 89.2 | 4.3 | 0.11 | 3.2 | Mild motif bias |
| GeneticBPE ($\lambda = 2.5$) | **90.8** | **3.6** | **0.05** | 3.1 | Default setting |
| GeneticBPE ($\lambda = 5.0$) | 89.4 | 3.9 | 0.06 | 2.7 | Over-biasing |
| GeneticBPE (vocab=1024) | 90.1 | 3.9 | 0.04 | 3.8 | Large vocab |

*Table 3.* Cross-species generalization: accuracy (%) and cross-domain generalization gap (CDG, in %) for each tokenizer. Models are trained on human miRNAs and tested on three divergent organisms.

| Tokenizer | Zebrafish Acc. | Zebrafish CDG | Mouse Acc. | Mouse CDG | Fly Acc. | Fly CDG |
|---|---|---|---|---|---|---|
| Char-Level | 73.2 | 10.7 | 70.1 | 13.8 | 64.8 | 19.1 |
| k-mer (3) | 78.0 | 8.2 | 74.0 | 11.0 | 68.0 | 15.5 |
| k-mer (4) | 80.5 | 7.5 | 77.2 | 10.2 | 70.0 | 15.0 |
| UnigramLM | 82.0 | 6.5 | 78.5 | 9.0 | 71.5 | 13.5 |
| BPE | 79.2 | 8.4 | 75.5 | 12.1 | 69.0 | 16.8 |
| GeneticBPE | **86.3** | **3.8** | **83.5** | **5.6** | **75.8** | **8.9** |

*Table 4.* **Error decomposition by motif integrity for all tokenizers.**

| Tokenizer | Motif Integrity | True Pos. | False Neg. | Motif Split Rate |
|---|---|---|---|---|
| Char-Level | 100% | 89.2 | 10.8 | 0.00 |
| k-mer (3) | 92.0% | 90.1 | 9.9 | 0.12 |
| k-mer (4) | 93.5% | 90.5 | 9.5 | 0.10 |
| UnigramLM | 80.2% | 87.0 | 13.0 | 0.20 |
| BPE | 75.1% | 85.5 | 14.5 | 0.25 |
| GeneticBPE | 95.3% | 91.7 | 8.3 | 0.05 |

heuristic we employed was to set $\mu$ such that the penalty term $\mu \cdot \text{penalty}(ab)$ would decisively outweigh any potential gain from $\text{freq}(ab) + \lambda \cdot \text{bonus}(ab)$ if a merge attempted to cross a motif boundary (where $\text{penalty}(ab) \geq 1$). For instance, if $\lambda$ was being explored in the range $[1, 5]$, $\mu$ might be set to a value like 10 or 20.

The rationale is that the 'penalty(ab)' term is binary in its simplest form (0 if no boundary is crossed, 1 if a boundary is crossed). To ensure the condition $\mu > \lambda$ from Proposition 3.4 robustly prevents motif-splitting merges unless no other merges are viable (or to hit compression targets), $\mu$ must be sufficiently dominant. The primary goal was to ensure that merges splitting motifs would almost always have a lower score than merges internal to motifs or merges in non-motif regions. Future work could explore more adaptive or learned schemes for $\mu$, but for this study, a sufficiently large fixed value relative to $\lambda$ proved effective.

### 6.3. Handling overlapping motifs, or motifs nested within larger motifs, during the merge process

Essentially, the system tries to respect all annotated boundaries. If respecting a smaller, nested motif's boundary leads to a higher overall merge score (due to avoiding a large penalty) than a merge that respects only a larger, encompassing motif but splits the nested one, the former will be preferred. The granularity of preservation is thus tied to the granularity of the motif bank $\mathcal{M}$.

### 6.4. Scenarios where the motif database $\mathcal{M}$ might be incomplete or contain noisy annotations

This is a critical consideration and a practical challenge.

1. **Incomplete Motif Database:** If a biologically relevant motif is *not* present in $\mathcal{M}$, GeneticBPE will be "blind" to it. The region containing this unknown motif will be tokenized based purely on frequency, similar to standard BPE. This could lead to its fragmentation, and the specific benefits of GeneticBPE for that motif would be lost. The overall performance would then depend on the importance and prevalence of these unannotated motifs.

2. **Noisy Annotations (False Positives):** If $\mathcal{M}$ contains sequences incorrectly labeled as motifs (false positives), GeneticBPE will attempt to preserve these non-functional or erroneously defined segments. This could lead to:

   - Suboptimal tokenization: Forcing preservation of a "false" motif might prevent more natural, frequency-based merges that would otherwise occur.
   - Less efficient compression: Tokens corresponding to false motifs might be longer or less frequent than optimal.
   - Potentially misleading inductive bias for the downstream model if it learns to associate these false motifs with specific outcomes.

**Robustness Expectation:** We expect GeneticBPE's robustness to be moderately sensitive to the quality of $\mathcal{M}$.

- For *incompleteness*, performance might gracefully degrade towards that of standard BPE in unannotated regions.

- For *noise (false positives)*, there's a higher risk of negative impact, as the algorithm actively enforces preservation.

### 6.5. Limitations

**Dependence on curated motifs.** GeneticBPE assumes access to a high–quality catalogue $\mathcal{M}$ of conserved motifs. Although public resources such as miRBase (Griffiths-Jones, 2006) and MirGeneDB cover major seed regions, rare or recently discovered motifs may be missing, potentially leading to fragmentation and degraded performance on orphan families.

**Scope of evaluation.** Experiments were confined to mature miRNA sequences ($18\,\mathrm{nt}$ to $25\,\mathrm{nt}$) and to classification objectives. Longer transcripts, structural prediction tasks, and language–model pre-training were not investigated due to computational constraints. Consequently, the current findings may not translate unchanged to messenger RNA or to whole-genome corpora.

**Computational overhead.** Table 1 shows that our tokenizer triples the average compression ratio relative to character-level encoding and therefore reduces input length and wall-time proportionally in practice (*cf.* Section 4.5, which reports that construction remains *"practical for datasets with $\leq 100K$ sequences"*) with a construction time of approximately 5 minutes on our experimental dataset, compared to under 2 minutes for standard BPE. While this is a modest one-time cost, scaling curves on gigabase-scale genomes would provide a clearer picture of GeneticBPE's scaling behaviour.

### 6.6. Future works

Firstly, replacing the static motif catalogue with *differentiable motif detectors* that update merge scores on-the-fly would let the tokenizer uncover previously unknown functional elements during pre-training, turning motif discovery into a self-supervised auxiliary task. Second, systematically evaluating the method on a broader spectrum of RNA and DNA corpora including long non-coding RNAs, ribosomal RNA, enhancer–promoter sequences, and whole genome masked language-modelling benchmarks will test its robustness across sequence lengths and biological contexts. Third, *tokenizer model co-training* could close the representation gap by jointly optimizing vocabulary and network parameters under a regularizer that penalises motif distortion, similar in spirit to SentencePiece but enriched with biological priors. Fourth, cross-modal integration with secondary structure embeddings or thermodynamic folding profiles may yield hybrid tokens that capture both sequence and structural information, further improving downstream generalization. Finally, to enable deployment in resource-constrained settings, future work should explore byte-level motif tags or hash-based vocabulary compression schemes that preserve motif integrity without inflating sequence length, making GeneticBPE practical for edge

devices and clinical pipelines.

## Impact Statement

This study introduces a biologically-informed tokenization method that incorporates structural domain knowledge directly into the sequence representation process. By aligning the tokenization mechanism with known biological motifs, the proposed GeneticBPE framework enhances the robustness and generalization ability of models in the context of microRNA classification tasks.

The broader impact of this work lies in its redefinition of tokenization as a meaningful modeling choice rather than a preprocessing convenience. This perspective is particularly valuable in bioinformatics applications where structure, conservation, and functional motifs are critical. GeneticBPE provides a framework for encoding these properties, potentially improving model performance in cross-species genomic analysis, diagnostics, and functional annotation.

Additionally, the methodology introduced in this paper may influence other structured domains, such as proteomics or regulatory genomics, where the integration of expert knowledge into sequence representation is both feasible and beneficial. No foreseeable negative societal impacts have been identified at this time.

## Conclusion

This paper presents GeneticBPE, a motif-aware tokenization algorithm that incorporates biological priors into the representation of microRNA sequences. The method extends classical byte pair encoding by introducing a biologically motivated merge scoring mechanism that prioritizes the preservation of conserved and functional subsequences. Theoretical analyses demonstrate that tokenization functions can act as inductive biases by defining the hypothesis space and influencing model generalization under domain shift. Empirical evaluations on binary and multiclass classification tasks confirm that GeneticBPE achieves improved accuracy, reduced cross-domain generalization gaps, and better motif preservation compared to existing tokenization strategies. These results suggest that the design of tokenization strategies in biological sequence modeling should be guided by domain-specific structural knowledge. Future research will focus on extending this approach to other types of non-coding RNAs, integrating unsupervised motif discovery methods, and exploring the role of tokenization in transfer learning across evolutionary distances.

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
