# OpenReview forum: "GeneticBPE: Motif-Preserving Tokenization for Robust miRNA Modeling"
_ICML.cc/2025/Workshop/TokShop — TokShop_

### Official Review · Reviewer_Ci23 · 2025-06-04
**A Motif-Preserving Tokenizer with Promising Potential and Structural Limitations in miRNA Modeling**

**Rating:** 7
**Confidence:** 4

**Review:**

The paper proposes a new tokenization method--GeneticBPE--designed to preserve biologically meaningful motify, specifically for miRNA seed regions, which are functionally important and highly conserved regions, during the tokenization of nucleotide sequences for transformer-based models. It introduces a merge score that rewards merges that stay within motifs and penalizes those that split across motif boundaries. This algorithm ensures that the motif distortion rate does not increase. The authors empirically evaluated accuracy, cross-species generalization, motif fidelity on miRNA classification tasks, and show superiority over k-mer, Unigram tokenization methods.

Strengths:
* Theoretically grounded, linking motif preservation.  The main contribution of this paper is that it explicitly integrates biologically validated motifs (e.g., miRNA seed regions) into the tokenization process, preventing harmful fragmentation.

* Strong empirical result. Achieved the highest classification accuracy on both binary (BurBary) and multiclass (MultiTop50) miRNA classification tasks and the lowest motif distortion rate among compressed tokenizers.

* Computationally feasible without too much increase in latency. GeneticBPE achieves high compression (~3.1× vs 1.0× for char-level) while preserving ~95% of motifs. This maintains computational efficiency without sacrificing biological fidelity.

Potential for improvement (weakness):
* The paper mentioned that they use an expert-annotated motif database source from miRBase and MirGeneDB v3.0, however, it would benefit from some quality control and description of the data composition. For instance, describe the exact number of unique motifs used?

* Currently, the evaluation is limited to miRNA classification; broader applications like ncRNA, siRNA etc. were not seen. It might be more generalizable to include those.

* Tokenization is performed separately and statically before model training. There’s no opportunity to adapt the tokenizer based on downstream task loss or model feedback, potentially leading to suboptimal token boundaries for the target task. You may consider integrating the tokenizer into an end-to-end differentiable pipeline and using model loss to guide token merges dynamically.

---

### Official Review · Reviewer_Q1SS · 2025-06-05
**Strong and interesting method for tokenization of genetic data with clear downstream results**

**Rating:** 8
**Confidence:** 3

**Review:**

This paper introduces GeneticBPE, a biologically-informed tokenization method that modifies standard BPE to preserve functional motifs in miRNA sequences. The key idea is to incorporate motif-preservation constraints into the merge scoring function, preventing fragmentation of biologically meaningful patterns.

Strengths:

-    The paper seems novel and strong, with practical relevance
-    Clear motivation, a nice simple idea that works
-    Good ablations and clear improvements in results

Weaknesses:

-    Section 3 is largely redundant, introducing mathematical notation whose purpose seems to only to impress (but is more likely to confuse). Most of it is never used, and statements that could be a single short sentence are presented as formulas. I suggest to integrate what little is used into section 4, and cut/appendicize the rest.
-    The literature references to related recent work on language tokenizers could be strenghtened (pretokenization, character/word boundaries, morphologically aware tokenization).
-    Are there relevant non-transformer baselines for these tasks to compare to?

Finally, one thing I wondered and will leave here in case it helps:
 -   Could the motifs be forcibly 'pre-tokenized' instead as an alternative to merge rules, similar to how words are split in NLP?

Definitely Top 50%, could be top 15% if weaknesses are addressed.

---

### Official Review · Reviewer_bjNn · 2025-06-09
**Motif-Aware Tokenization Enhances miRNA Model Robustness and Generalization**

**Rating:** 7
**Confidence:** 4

**Review:**

This paper introduces GeneticBPE, a biologically informed tokenization algorithm designed to preserve structural motifs in miRNA sequences during tokenization. Traditional methods like BPE or Unigram focus on frequency-based merges and often ignore biologically meaningful patterns, resulting in fragmented motifs and degraded performance under domain shift (e.g., cross-species). GeneticBPE addresses this by incorporating domain-specific constraints that reward motif-internal merges and penalize motif-splitting ones.

Strengths:
- The paper makes a compelling theoretical case for treating tokenization not as a preprocessing step but as a model-aligned component that influences generalization, especially in structured biological domains.
- GeneticBPE outperforms competing tokenizers on both binary and multiclass miRNA classification tasks. It achieves the highest accuracy and lowest generalization gap across all evaluated species.
- Formal definitions and bounds (e.g., motif distortion rate, domain-shift-aware generalization bound) provide rigorous support for the method’s design.

Weaknesses:
- The merge score formula (freq + λ·bonus − μ·penalty) does not normalize for the relative size of the motif dataset versus the sequence corpus. As corpus size increases, the frequency term may dominate, making it increasingly difficult to select λ and μ values that generalize well. This introduces a new hyperparameter tuning challenge, or even a surrogate machine learning problem, for every new dataset scale.
- The approach depends heavily on a high-quality curated motif set. Its performance may degrade with incomplete or noisy annotations, a limitation partially acknowledged but not experimentally explored.
- Evaluation is limited to short miRNA sequences and classification objectives. It is unclear how well the method generalizes to longer sequences or other tasks like structural prediction or generative modeling.
- The paper does not report variance or confidence intervals across different random seeds for downstream evaluations. Without such reporting, it’s unclear whether the observed gains are robust or sensitive to specific initialization settings.

---

### Decision · Program_Chairs · 2025-06-10

Accept